# Correlation between Visual Acuity and Optical Coherence Tomography Angiography Parameters in Unilateral Idiopathic Epiretinal Membrane

**DOI:** 10.3390/jcm10010026

**Published:** 2020-12-24

**Authors:** I-Mo Fang, Hsin-Yi Hsu, Wan-Ling Chiang, Yi-Ling Shih, Chia-Ling Han

**Affiliations:** 1Department of Ophthalmology, Taipei City Hospital, Zhongxiao Branch, No. 87, Tonde Road, Nankang District, Taipei 10002, Taiwan; B1887@tpech.gov.tw (H.-Y.H.); a4383@tpech.gov.tw (W.-L.C.); a0919480206@gmail.com (C.-L.H.); 2Department of Ophthalmology, National Taiwan University Hospital, Taipei 11556, Taiwan; cgineline@gmail.com; 3Department of Special Education, University of Taipei, Taipei 11153, Taiwan

**Keywords:** idiopathic epiretinal membrane (iERM), optical coherence tomography angiography (OCTA), visual acuity, choroidal capillary plexus (DCP), superficial capillary plexus (SCP), deep capillary plexus (DCP)

## Abstract

Background: The tangential traction by idiopathic epiretinal membrane (iERM) may alter the hemodynamics of the macula. We investigated the correlation between visual acuity and the optical coherence tomography angiography (OCTA) parameters in unilateral iERM. Methods: We included 61 eyes of 61 consecutive patients with unilateral iERM between January 2018 and December 2018. The flow area of the retinal superficial capillary plexus (SCP), deep capillary plexus (DCP), and choroidal capillary plexus (CCP) were measured using OCTA. The normal fellow eyes were used for comparison. The iERM patients were divided into those with a presence of foveal concavity and those with a loss of foveal concavity. Results: When compared with fellow eyes, the flow areas showed a statistically significant decrease in the SCP and CCP of those with iERM (*p* = 0.037 and *p* = 0.011, respectively). In the DCP, no significant reduction in flow area was found in iERM (*p* = 0.054). The flow area of the CCP was the only factor significantly associated with best vision (*p* = 0.012). No significant differences in the flow areas of the SCP, DCP, and CCP were found between the presence and loss of foveal concavity. Conclusions: The flow area of the CCP is an important determinant of vision, emphasizing the crucial role of choroidal circulation in iERM. Moreover, mechanical stretch by iERM is not the only mechanism affecting the flow area.

## 1. Introduction

Idiopathic epiretinal membrane (iERM) is a common macular disease, occurring in approximately 4% to 10% of the general population, and its prevalence increases with age [1,2]. The contractile properties of iERM may exert traction on the retina that may cause variable loss of visual acuity and metamorphopsia [3,4,5]. The tangential traction by iERM may cause retinal folds and the excursion or tortuosity of retinal vessels, which may alter the hemodynamics of the macula [6,7,8]. However, there is limited literature on the association between macular microvascular changes and visual acuity in iERM [9].

Regarding the factors affecting vision in patients with iERM, most studies have focused on the structural changes in the retina caused by iERM [10]. Disruption of the inner/outer segment junction of the photoreceptor layer is usually considered a sign of poor vision [11,12]. However, these macular structural changes are not always related to the degree of vision loss [13,14,15]. Optical coherence tomography angiography (OCTA) is a novel imaging modality that allows evaluation of the retinal and anterior choroidal vascular circulations without the need for dye injection [16,17,18]. OCTA enables closer observation of blood flow in retina and choroid. OCTA has been used many times to study the changes in macular microvascular characteristics and visual prognosis after macular surgery, highlighting the importance of macular microcirculation to visual acuity in iERM [19,20].

In this study, we investigated the changes in macular microvessels and compared them with normal fellow eyes in patients with unilateral iERM using OCTA. We explored the correlation between the best-corrected visual acuity and the OCTA parameters in these patients.

## 2. Materials and Methods

We conducted a prospective study of 61 consecutive patients who were diagnosed with idiopathic ERM (iERM) from 1 January 2018 to 31 December 2018 in the Department of Ophthalmology at Taipei City Hospital, Zhongxiao branch, Taiwan. The study design was approved by the Institutional Review Board of Taipei City Hospital (approval TCHIRB-10801013-E), and the study adhered to the tenets of the Declaration of Helsinki. iERM was diagnosed based on the presence of a fibrous membrane in front of the macula, as visible on fundus examination using a 90 dpt lens, and was confirmed with Optical coherence tomography (OCT). Patients were excluded if they had eyes with secondary ERM (diabetic retinopathy, venous occlusion, retinal tear, retinal detachment, uveitis, trauma, etc.), eyes with myopia of >6 diopters, or eyes with other ocular pathologic features that could have interfered with the functional results, such as glaucoma, visually significant cataract, or age-related macular degeneration. The fellow eyes without iERM in these patients served as normal controls.

All patients underwent thorough ophthalmologic examinations, including best-corrected visual acuity (BCVA; Snellen visual acuity chart), biomicroscopy of the anterior and posterior segments, indirect ophthalmoscopy, fundus photography, and OCT-A (RTVue XR Avanti, OptoVue, Inc., Fremont, CA, USA). All eyes were dilated with 1% tropicamide eye drops before OCT-A scanning. This apparatus has an A-scan rate of 70,000 scans per second and a B-scan frame rate of approximately 200 frames per second, using a light source with a bandwidth of 45 nm centered on 840 nm. Individual scans of layers of the retina and choroid were collected as A-scans. A-scans were compiled into a B-scan. Subjects with poor image quality were excluded based on the presence of one or more of the following criteria: low signal strength index (SSI < 50), the presence of one or more blink artifacts, poor fixation leading to motion or doubling artifacts, and media opacity obscuring the view of the vasculature. The splitspectrum amplitude decorrelation angiography (SSADA) algorithm compared the amplitude fluctuation of the reflected light between the consecutive B-scans at different locations so that the static tissue had a low decorrelation value, whereas blood flow had a high decorrelation value. Then, blood flow could be distinguished from static background tissue. Each subject in this study underwent three repeated B-scans at the same position of a 6 × 6 mm^2^ area of scanning, which was centered on the fovea. We analyzed the layers of the superficial capillary plexus (SCP), deep capillary plexus (DCP) and choroidal capillary plexus (CCP) from the en face OCT angiogram. The flow area refers to a circular region centered on the fovea with a radius of 2.0 mm, calculated by the number of pixels over the threshold (Figure 1). The SCP slab was taken from the internal limiting membrane to the inner plexiform layer/inner nuclear layer interface, and the DCP slab was taken from the inner plexiform layer/inner nuclear layer interface to the outer plexiform layer/outer nuclear layer interface.

### 2.1. Average Macular Thickness Assessment

The retina was automatically segmented into inner and outer portions, and the mean macular thickness in the fovea (within a circle of 1 mm diameter) and the parafovea (within a circle of 3 mm diameter) was calculated. In our study, the average macular thickness was defined as the mean macular thickness within a 3 mm diameter and was calculated using the formula: average macular thickness = 1/3 foveal thickness +2/3 parafoveal thickness.

### 2.2. Statistical Analysis

Data on the blood flow areas of four layers, the BCVA, and the macular thickness were recorded by two independent technicians. All data are presented as the means ± standard deviations (SDs). A paired *t*-test was used for comparison between iERM and normal fellow eyes. An unpaired *t*-test was used for comparison between the presence and loss of foveal concavity in iERM eyes. Pearson correlation analysis was used to analyze the associations between the OCTA parameters, where *p* < 0.05 was regarded as statistically significant.

## 3. Results

In this study, 61 eyes of 61 consecutive patients with a unilateral ERM diagnosed by using OCTA were included. The patients’ average age was 66.67 ± 8.88 years (range, 45 to 88 years); 21 patients (34.5%) were female. The spherical equilibrium (SE) was −2.12 ± 0.76 diopter (range, +2.25 to −5.25) in iERM eyes and −2.25 ± 0.83 diopter (range +1.75 to −5.0) in normal fellow eyes. There was no statistically significant difference between iERM eyes and normal fellow eyes in SE (*p* = 0.76, paired *t*-test). The average LogMAR BCVA (Logarithm of the Minimum Angle of Resolution) was 0.23 ± 0.22 (Snellen 20/34 equivalent) in iERM eyes and 0.10 ± 0.11 (Snellen 20/25 equivalent) in normal fellow eyes. The BCVA of iERM eyes was, statistically significantly worse than normal fellow eyes (*p* = 0.001, paired *t*-test).

Table 1 shows the differences in the OCTA parameters between iERM and normal fellow eyes. The flow area of SCP and CCP in iERM eyes was significantly lower than that of normal fellow eyes (*p* = 0.037 for SCP and *p* = 0.011 for CCP, paired *t*-test). The flow area of DCP in iERM eyes did not reach statistically significant differences compared with normal fellow eyes (*p* = 0.054, paired *t*-test). The thickness of the inner retina, outer retina, and total retina in iERM eyes increased significantly compared with normal fellow eyes (all *p* < 0.001, paired *t*-test). The mean choroidal thickness of iERM eyes also showed a statistically significant decrease (*p* = 0.042, paired *t*-test).

In iERM eyes, the OCTA parameters measured in the CCP had a significant negative correlation with LLogMAR BCVA (r = −0.33, *p* = 0.012, Pearson correlation test), but no significant correlation with LogMAR BCVA was found in the SCP and DCP (r = −0.22, *p* = 0.095 for SCP; r = −0.21, *p* = 0.11 for DCP; Pearson correlation test) (Table 2). The inner retinal, outer retinal, full retinal, and choroidal thicknesses were not significantly correlated with LogMAR BCVA in iERM eyes (r = 0.039, *p* = 0.75 for inner retina; r = 0.038, *p* = 0.78 for outer retina; r = 0.163, *p* = 0.23 for full retina; r = 0.069, *p* = 0.62 for choroidal thickness; Pearson correlation test). Figure 2 shows scatter plots of the OCTA parameters, such as the flow area of the SCP, DCP, and CCP, the inner retinal, outer retinal, full retinal and choroidal thicknesses, and LogMAR in iERM and normal fellow eyes. For normal fellow eyes, no significant association was identified between the flow area of the SCP, DCP, and CCP and LogMAR BCVA (r = −0.13, *p* = 0.45 for SCP; r = −0.053, *p* = 0.75 for DCP; r = −0.24, *p* = 0.16 for CCP; Pearson correlation test). Similarity, the inner retinal, outer retinal, total retinal, and choroidal thicknesses were not significantly correlated with LogMAR BCVA in normal fellow eyes (r = 0.075, *p* = 0.68 for inner retina; r = 0.17, *p* = 0.35 for outer retina; r = 0.13, *p* = 0.46 for full retina; r = 0.26, *p* = 0.12 for choroidal thickness; Pearson correlation test).

To investigate the influence of the foveal contour on the LogMAR BCVA and OCTA parameters in iERM patients, we subdivided iERM eyes into two subgroups: those with the presence of foveal concavity according to OCT morphology, and those with a loss of foveal concavity. Of the 61 eyes, 32 (52.46%) had foveal concavity, and 29 had a loss of foveal concavity. There were no statistical differences in age, gender, SE, or LogMAR BCVA between the two subgroups (*p* = 0.13 for age; *p* = 0.82 for gender; *p* = 0.73 for SE; *p* = 0.31 for LogMAR BCVA; unpaired *t*-test). Eyes with a loss of foveal concavity were significantly thicker in the inner retina, outer retina, and full retina than those with foveal concavity (*p* = 0.025 for inner retina; <0.001 for outer retina; <0.001 for full retina, unpaired *t* test). However, no statistical differences in the flow area in the SCP, DCP, and CCP were found between the two subgroups (*p* = 0.30 for SCP; *p* = 0.67 for DCP, *p* = 0.54 for CCP; unpaired *t*-test). There were also no statistical differences in the choroidal thickness between the two subgroups (*p* = 0.17 for choroidal thickness; unpaired *t*-test) (Table 3).

## 4. Discussion

We tested unilateral iERM patients and used fellow healthy eye as controls to reduce human variation errors. Moreover, iERM eyes with myopia less than 6 diopters were recruited to reduce the effect of the length of the eye axis on the blood flow of the retina and choroid. We found that compared with normal control eyes, the flow areas of iERM eyes in the SCP and CCP were significantly reduced. There was no statistically significant change in the flow area in the DCP. Furthermore, we found that only the flow density of the CCP layer was significantly related to corrected vision. Our results provide evidence that macular choroidal microcirculation in ERM patients is an important factor affecting vision.

We found that the flow area of the SCP and CCP in ERM eyes had a statistically significant decrease, while that of the DCP had a marginally significant decrease compared with normal fellow eyes. It is generally believed that ERM distributed in the macula can exert a mechanical force on the superficial retinal vessels, primarily affecting the SCP vessels, and may cause proportionate impairment in the DCP and CCP [21,22,23]. However, we demonstrated that the flow area of the CCP, which theoretically suffered the least tractional force by ERM, was significantly reduced, while the flow area of the DCP was less affected. Chen et al. also showed that iERM significantly decreases the mean foveal vessel density of CCP, which is reversible by surgery [24]. Our findings imply that mechanical stretch by iERM is not the only mechanism affecting the flow area of iERM patients. Several possible explanations have been proposed, including that mechanical stretching of retinal pigment epithelium(RPE) may increase the level of vascular endotheilal growth factor (VEGF), leading to increased retinal and choroidal vascular permeability, and that local low-grade inflammation induced by ERM may be implicated in vascular changes [25,26]. Further large-scale and extensive studies are needed to clarify this point.

We found that, compared with eyes with foveal concavity, eyes with a loss of foveal concavity had increased thickness of the inner and outer retina, but the flow area did not change in the SCP, DCP, or CCP layers, and there was no significant effect on best-corrected vision. Our results agree with previous studies that demonstrated that foveal contour is not significantly associated with BCVA [27]. The contour of the fovea is a useful indicator of traction on the retina; the traction resulted in an elevation where the foveal depression should be located. Therefore, our results showed that there was no significant difference in the flow area of the SCP, DCP, and CCP layers between iERM eyes with and without foveal depression, indicating again that the traction is not the only factor affecting the flow areas of iERM patients.

Several studies have indicated that choroidal thickness is associated with visual acuity [28,29]. Nishida et al. found that subfoveal choroidal thickness is the only significant predictor of visual acuity in highly myopic eyes. They suggested that the thin choroid of highly myopic eyes may deliver decreased amounts of nutrients and oxygen to the outer retina, which may consequently affect the function and structures of the photoreceptors [30,31,32,33]. In our study, we extended the scope and found that macular flow density of choriocapillaris was the only significant factor correlated with visual acuity in iERM eyes. In iERM eyes, we found that the choroidal thickness and CCP in the macular area were decreased in comparison to those of normal fellow eyes. Therefore, the reason why the visual acuity of iERM eyes was related to choroid is similar to that of highly myopic eyes. However, although there was a high correlation between choroidal blood flow and choroidal thickness, our results showing that vision is only related to the macular flow area of the CCP and not to choroidal thickness, which were different from previous studies.We believe that this discrepancy was because we were measuring the macular choroidal thickness instead of the subfoveal thickness measured in previous studies. Further study is needed to clarify this point by measuring only the subfoveal choroidal thickness and flow area.

There were several limitations in this study. Firstly, it used a relatively small sample size, which might have yielded false-negative or false-positive results. Further studies with larger sample sizes are warranted to check the accuracy of our findings. Secondly, we excluded the cases with poor signal strength. Some of these were due to severe retinal distortion and segmentation errors, which could have confounded the measurements and OCTA signals. Thus, it is possible that we might have induced a selection bias by excluding the severe iERM cases.

## 5. Conclusions

In conclusion, we found that the flow area of the CCP in OCTA is an important determinant of vision in iERM patients, emphasizing the crucial role of choroidal circulation in iERM. Moreover, mechanical stretch by iERM is not the only mechanism affecting the flow areas of iERM patients. Our results may provide evidence of the influence and mechanisms of iERM on retinal and choroidal microcirculation.

## Figures and Tables

**Figure 1 jcm-10-00026-f001:**
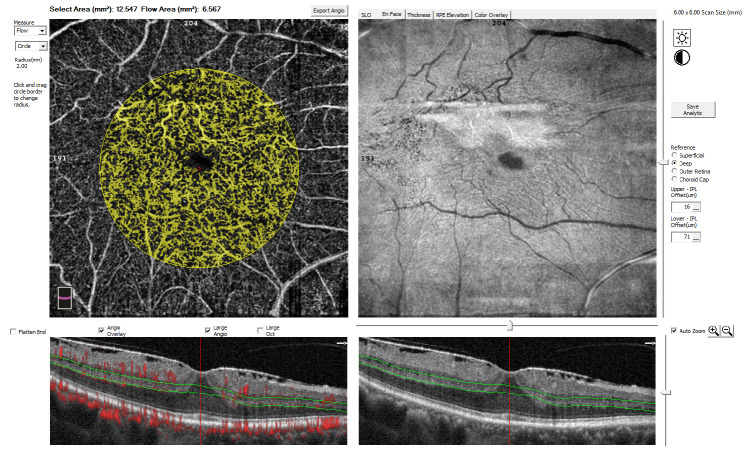
Example of the flow area measured by using optical coherence tomography angiography (OCTA). The flow area (highlighted in yellow) is marked as a circle, centered on the fovea, with a radius of 2.0 mm. In this example, the deep capillary plexus (DCP), delineated by green lines, was analyzed. The flow area was measured automatically as 6.567 mm^2^.

**Figure 2 jcm-10-00026-f002:**
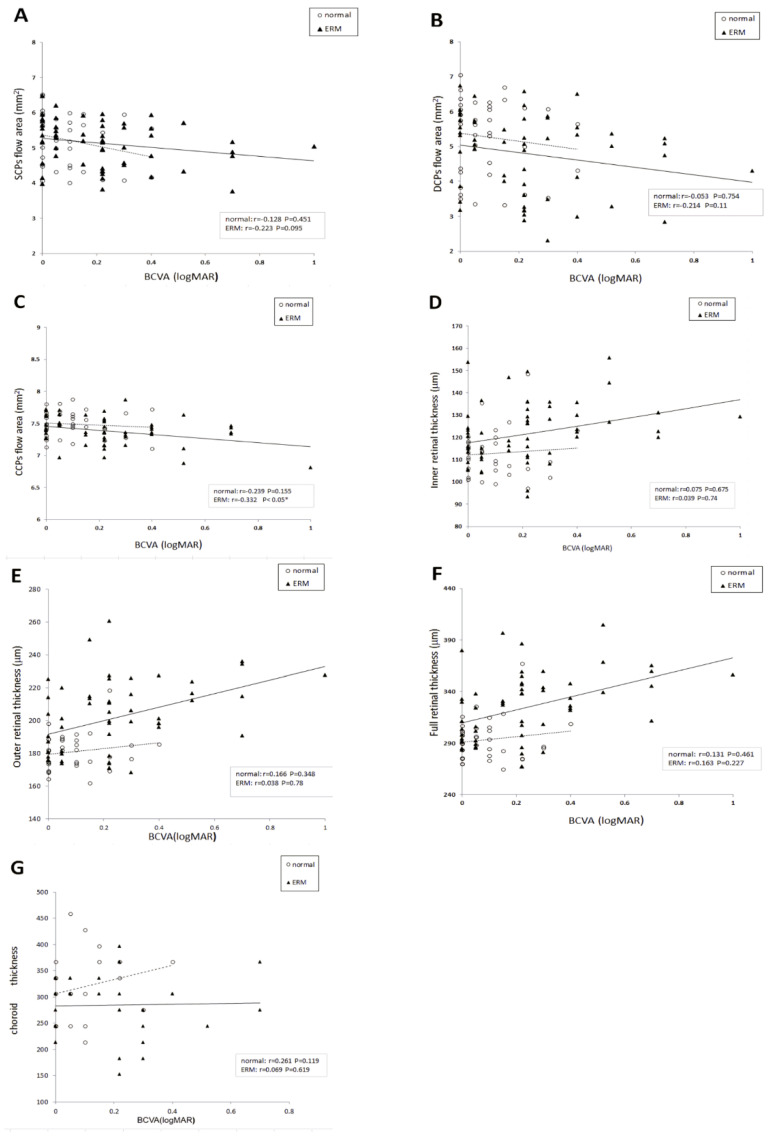
Scatter plots showing associations between best-corrected visual acuity (BCVA) and the flow areas of the retinal (**A**) superficial capillary plexus (SCP), (**B**) deep capillary plexus (DCP), and (**C**) choroidal capillary plexus (CCP), as well as the thickness of the (**D**) inner retina, (**E**) outer retina, (**F**) full retina, and (**G**) choroid, as measured by using optical coherence tomography angiography (OCTA). Pearson’s correlation coefficient (r) and *p*-values for the slope of the regression line are noted.

**Table 1 jcm-10-00026-t001:** OCTA parameters in iERM eyes and normal fellow eyes.

	iERM (*n* = 61)	Normal (*n* = 61)	*p*
SE (diopter)	−2.12 ± 0.76	−2.25 ± 0.83	0.76
BCVA (LogMAR)	0.23 ± 0.22	0.10 ± 0.11	0.001 *
SCP flow area (mm^2^)	5.006 ± 0.501	5.28 ± 0.75	0.037 *
DCP flow area (mm^2^)	4.90 ± 1.11	5.26 ± 1.08	0.054
CCP flow area (mm^2^)	7.287 ± 0.214	7.663 ± 0.201	0.011 *
Inner retinal thickness (μm)	324.87 ± 32.01	112.72 ± 10.66	<0.001 *
Outer retinal thickness (μm)	201.91 ± 21.84	180.95 ± 10.86	<0.001 *
Full retinal thickness (μm)	324.87 ± 32.01	293.48 ± 20.11	<0.001 *
Choroid thickness (μm)	270.79 ± 59.61	300.05 ± 76.53	0.042 *

All data are presented as the means ± SDs. OCTA, optical coherence tomography angiography; iERM, idiopathic epiretinal membrane; SE, spherical equilibrium; BCVA, best-corrected visual acuity; LogMAR, Logarithm of the minimum angle of resolution; SCP, superficial capillary plexus; DCP, deep capillary plexus; CCP, choroidal capillary plexus. * The value was statistically significant (*p* < 0.05).

**Table 2 jcm-10-00026-t002:** Pearson correlation analysis of BCVA and OCTA parameters in iERM and normal fellow eyes.

	BCVA (LogMAR)			
	iERM		Normal Fellow	
	r	*p*	r	*p*
SCP flow area	−0.22	0.095	−0.13	0.45
DCP flow area	−0.21	0.11	−0.053	0.75
CCP flow area	−0.33	0.012 *	−0.24	0.16
Inner retinal thickness	0.039	0.74	0.075	0.68
Outer retinal thickness	0.038	0.78	0.17	0.35
Full retinal thickness	0.16	0.23	0.13	0.46
Choroid thickness	0.069	0.62	0.26	0.12

BCVA, best-corrected visual acuity; LogMAR, Logarithm of the minimum angle of resolution; OCTA, optical coherence tomography angiography; iERM, idiopathic epiretinal membrane; SCP, superficial capillary plexus; DCP, deep capillary plexus; CCP choroidal capillary plexus; r, Pearson r value. * The value was statistically significant (*p* < 0.05).

**Table 3 jcm-10-00026-t003:** OCTA parameters in iERM eyes with and without the presence of foveal concavity.

	Presence of Foveal Concavity	Loss of Foveal Concavity	*p*
Number of eyes	32	29	
age (years)	67.76 ± 8.65	63.82 ± 8.21	0.13
gender (female; %)	0.29 ± 0.46	0.32 ± 0.48	0.82
SE (diopter)	2.28 ± 0.86	2.06 ± 0.93	0.73
BCVA (LogMAR)	0.20 ± 0.16	0.26 ± 0.28	0.31
SCP flow area (mm^2^)	5.28 ± 0.64	5.03 ± 0.63	0.30
DCP flow area (mm^2^)	4.90 ± 1.16	4.77 ± 1.01	0.67
CCP flow area (mm^2^)	7.41 ± 0.21	7.41 ± 0.20	0.54
Inner retinal thickness (μm)	126.26 ± 14.28	119.25 ± 8.39	0.025 *
Outer retinal thickness (μm)	203.63 ± 22.73	184.36 ± 17.03	<0.001 *
Full retinal thickness (μm)	330.86 ± 34.42	301.91 ± 21.26	<0.001 *
Choroid thickness (μm)	261.68 ± 60.91	283.68 ± 54.77	0.167

All data are presented as the means ± SDs. OCTA, optical coherence tomography angiography; iERM, idiopathic epiretinal membrane; SE, spherical equilibrium; BCVA, best-corrected visual acuity; LogMAR, Logarithm of the minimum angle of resolution; SCP, superficial capillary plexus; DCP, deep capillary plexus; CCP, choroidal capillary plexus. * The value was statistically significant (*p* < 0.05).

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
