# Peer review of "Correlation between Visual Acuity and Optical Coherence Tomography Angiography Parameters in Unilateral Idiopathic Epiretinal Membrane"

_jcm, 2020, doi:10.3390/jcm10010026_

Round 1

Reviewer 1 Report

The authors aimed to identify OCTA biomarkers that correlated with visual acuity in patient with idiopathic ERMs

Unfortunately, multiple methodological issues limit the validity of the results and their interpretation.

The most striking being the fact that with the currently available OCTA technology, the CCP is significantly affected by projection artifacts. This is further increased if an ERM is present.

The authors appear to completely omit any mention or discussion in this regard.

Also, as cited by the authors Chen et al., also showed that iERM significantly decreased the mean foveal vessel density of CCP, which is reversible by surgery. This is once again a suggestion that measurement bias may play an important role, and therefore no reliable conclusions can be made with these kind of measurements, unfortunately.

Author Response

Dear reviewer:

Thank you for the comprehensive review and valuable comments on our manuscript entitled “Correlation between Visual Acuity and Optical Coherence Tomography Angiography Parameters in Unilateral Idiopathic Epiretinal Membrane ‘’. We have revised the paper in accordance with the reviewers’ suggestions. In the following pages we will address each comment one by one. We appreciate the comments from the reviewers to improve the quality of our article. We hope our paper will be acceptable to be published in “Journal of Clinical Medicine ’’.  

Respectfully yours,

I-Mo Fang, M.D, Ph.D.

TEL: +886-2-27861288 ext.8275

FAX: +886-2-27888492

Point 1: The most striking being the fact that with the currently available OCTA technology, the CCP is significantly affected by projection artifacts. This is further increased if an ERM is present. The authors appear to completely omit any mention or discussion in this regard.

Response 1: We totally agree with the reviewer’s opinion. The currently available OCTA technology, the CCP is significantly affected by projection artifacts. This is further increased if an ERM is present. We noticed this type of bias and described it in the manuscript as a limitation of article design. In line 217, we described that “we excluded the cases with poor signal strength. Some cases with poor signal strength were due to severe retinal distortion and segmentation errors, which could confound measurements and OCTA signals. Thus, it is possible that we might induce a selection bias in excluding the severe iERM cases.”

Reviewer 2 Report

Fang et al. produced a very well-written article consisting of several experiments highlighting “Correlation between Visual Acuity and Optical Coherence Tomography Angiography Parameters in Unilateral Idiopathic Epiretinal Membrane”. I consider the manuscript very fascinating but, in the same time, I suggest several revisions needed to improve the quality and the readability of the paper:

  • Line 38: in “Introduction” the authors described association between macular microvascular changes and visual acuity. I suggest to update references adding the recent PMID: 33233546, describing the influence of angiogenesis in eye dystrophies.
  • Line 40-41: the authors said the “Disruption of the inner segment/ outer segment of the photoreceptor layer is usually considered a sign of poor vision”. I agree with the authors, and I suggest to improve the bibliography of the period, adding more detailed references, such as PMID: 28764803.
  • “Materials and Methods”: Did the authors realize all experiments at least in triplicate?
  • “Statistical Analysis” section lacks of post-hoc correction method (eg. FDR, Bonferroni or Tukey post-hoc test).
  • Line 201-203: in “Discussion” section, the authors said: “They suggested that the thin choroid of highly myopic eyes may deliver decreased amounts of nutrients and oxygen to the outers retina, which may consequently affect the function and structures of the photoreceptors”. Alterations linked to oxygenation impairments and oxidative stress are recently studied and evaluated in retinal cells. Several papers already published could be used as reference, involving also eye-related inflammatory diseases. Regarding these, I suggest to add the following references to manuscript PMID: 32290199, PMID: 32413970, PMID: 32326576 and PMID: 33233726.
  • Finally, manuscript requires English revisions and typos correction.
  • These references could be cited: PMID: 30569234, PMID: 25317632 and PMID: 26694492

Author Response

Dear reviewer:

Thank you for the comprehensive review and valuable comments on our manuscript entitled “Correlation between Visual Acuity and Optical Coherence Tomography Angiography Parameters in Unilateral Idiopathic Epiretinal Membrane ‘’. We have revised the paper in accordance with the reviewers’ suggestions. In the following pages we will address each comment one by one. We appreciate the comments from the reviewers to improve the quality of our article. We hope our paper will be acceptable to be published in “Journal of Clinical Medicine ’’.  

Respectfully yours,

I-Mo Fang, M.D, Ph.D.

TEL: +886-2-27861288 ext.8275

FAX: +886-2-27888492

Point 1 :Line 38: in “Introduction” the authors described association between macular microvascular changes and visual acuity. I suggest to update references adding the recent PMID: 33233546, describing the influence of angiogenesis in eye dystrophies.

Response 1: we have added the excellent reference PMID: 33233546 in our manuscript.

Point 2 : Line 40-41: the authors said the “Disruption of the inner segment/ outer segment of the photoreceptor layer is usually considered a sign of poor vision”. I agree with the authors, and I suggest to improve the bibliography of the period, adding more detailed references, such as PMID: 28764803.

Response 2: we have added the excellent reference PMID: 28764803 in our manuscript according to the suggestion of the reviewer.

Point 3:“Materials and Methods”: Did the authors realize all experiments at least in triplicate?

Response 3: Yes, each subject in this study underwent three repeated B-scans at the same position of a 6 mm × 6 mm area of scanning, which was centered on the fovea. We had corrected the mistake in line 76.

Point 4: Statistical Analysis” section lacks of post-hoc correction method (eg. FDR, Bonferroni or Tukey post-hoc test).

Reponse 4: Thank you for your suggestion. In this manuscript, we only use two variables paired or unpaired t test for statistics analysis. It seems that there is no need to use post hoc correction method.

Point 5: Line 201-203: in “Discussion” section, the authors said: “They suggested that the thin choroid of highly myopic eyes may deliver decreased amounts of nutrients and oxygen to the outers retina, which may consequently affect the function and structures of the photoreceptors”. Alterations linked to oxygenation impairments and oxidative stress are recently studied and evaluated in retinal cells. Several papers already published could be used as reference, involving also eye-related inflammatory diseases. Regarding these, I suggest to add the following references to manuscript PMID: 32290199, PMID: 32413970, PMID: 32326576

Reponse 5: we have added the excellent reference PMID: 32290199, PMID: 32413970, PMID: 32326576 in our manuscript.

Point 6: Finally, manuscript requires English revisions and typos correction.

Reponse 6: We have made English revisions and typos corrections. Thank you.

Reviewer 3 Report

This manuscript by Fang et al., concentrated on the investigation of correlation between visual acuity and OCTA parameters in unilateral iERM. The authors investigated the changes in macular microvessels and compared them with normal fellow eyes from 61 patients with unilateral iERM using advanced OCTA technique. It appears the data are of high quality and solid. There are several errors need to be corrected before acceptance.

  1. From line 15 to line 16, the authors stated “We included 61 eyes of 61 consecutive patients with unilateral iERM between Jan. 2018 and Dec. 2018.” However, from line 52 to line 53, the authors stated “We conducted a prospective of 61 consecutive patients who were diagnosed with idiopathic ERM (iERM) from March 1, 2019 to Feb 28, 2020 in the Department of Ophthalmology”, so the time discrepancy needs to be consistent.
  2. It appears that some numbers in “Table 2 (line 139-140)” and “Table 3 (line 162-163)” are missing some parts or missing some contents, please check it and correct it accordingly.
  3. From line 212, “Further study is need to clarify this point by measuring only the subfoveal choroidal”. Please change “need” to “needed”.
  4. From line 215, please change “false-native” to “false negative”.

Author Response

Dear reviewer:

Thank you for the comprehensive review and valuable comments on our manuscript entitled “Correlation between Visual Acuity and Optical Coherence Tomography Angiography Parameters in Unilateral Idiopathic Epiretinal Membrane ‘’. We have revised the paper in accordance with the reviewers’ suggestions. In the following pages we will address each comment one by one. We appreciate the comments from the reviewers to improve the quality of our article. We hope our paper will be acceptable to be published in “Journal of Clinical Medicine ’’.  

Respectfully yours,

I-Mo Fang, M.D, Ph.D.

TEL: +886-2-27861288 ext.8275

FAX: +886-2-27888492

Point 1: From line 15 to line 16, the authors stated “We included 61 eyes of 61 consecutive patients with unilateral iERM between Jan. 2018 and Dec. 2018.” However, from line 52 to line 53, the authors stated “We conducted a prospective of 61 consecutive patients who were diagnosed with idiopathic ERM (iERM) from March 1, 2019 to Feb 28, 2020 in the Department of Ophthalmology”, so the time discrepancy needs to be consistent.

Response 1: We apologize for the mistake. We conducted a prospective of 61 consecutive patients who were diagnosed with idiopathic ERM (iERM) from Jan. 1, 2018 to Dec. 31, 2018 in the Department of Ophthalmology. We have revised. Thank you.

Point 2:It appears that some numbers in “Table 2 (line 139-140)” and “Table 3 (line 162-163)” are missing some parts or missing some contents, please check it and correct it accordingly.

Response 2: We have corrected the mistakes. Thank you.

Point 3:From line 212, “Further study is need to clarify this point by measuring only the subfoveal choroidal”. Please change “need” to “needed”.

Response 3: We have corrected it. Thank you.

Point 4:From line 215, please change “false-native” to “false negative”.

Response 4: We have corrected it. Thank you.

Round 2

Reviewer 2 Report

The authors addressed all suggested points. Thanks.